# Epidemiology of Secondary Warm Autoimmune Haemolytic Anaemia—A Systematic Review and Meta-Analysis

**DOI:** 10.3390/jcm10061244

**Published:** 2021-03-17

**Authors:** Stinne Tranekær, Dennis Lund Hansen, Henrik Frederiksen

**Affiliations:** 1Haematological Research Unit, Department of Clinical Research, University of Southern, 5230 Odense M, Denmark; sttra16@student.sdu.dk (S.T.); dennis.lund.hansen2@rsyd.dk (D.L.H.); 2Department of Haematology, Odense University Hospital, 5000 Odense C, Denmark

**Keywords:** autoimmune haemolytic anaemia, epidemiology, autoantibodies, secondary causes, meta-analysis, review

## Abstract

Background: Warm autoimmune haemolytic anaemia (wAIHA) is a haemolytic disorder, most commonly seen among adults and is classified as either primary or secondary to an underlying disease. We describe the age and sex distribution and the proportion of secondary wAIHA. Method: We retrieved 2635 published articles, screened abstracts and titles, and identified 27 articles eligible for full-text review. From these studies, we extracted data regarding number of patients, sex distribution, age at diagnosis, number of patients with secondary wAIHA, and whether the patients were diagnosed through local or referral centres. All data were weighted according to the number of included patients in each study. Results: 27 studies including a total of 4311 patients with wAIHA, of which 66% were females, were included. The median age at diagnosis was 68.7 years, however, wAIHA affected all ages. The mean proportion of secondary wAIHA was 49%, most frequently secondary to systemic lupus erythematosus. The proportions of secondary wAIHA reported from primary vs. referral centres were 35% vs. 59%, respectively. Conclusion: This review consolidates previously reported gender distribution. The higher proportion of secondary wAIHA in referral centres suggests that the most severely affected patients are disproportionally more frequent in such facilities.

## 1. Introduction

Autoimmune haemolytic anaemia (AIHA) is an acquired haemolytic disorder where antibodies directed against red blood cell (RBC) surface epitopes, cause premature RBC destruction, which leads to anaemia [1,2,3]. Although rare, AIHA is one of the most common forms of acquired haemolytic disorders with an incidence rate of 1.8 pr. 100,000 person-years, and a prevalence of 9.5 per 100,000 [4]. AIHA can occur at any age, but is most common among older adults, and the majority of AIHA patients are above 40 years at diagnosis [4,5]. Females are reported to have an increased risk of developing AIHA, often explained by an increased prevalence of other autoimmune diseases commonly associated with AIHA, such as systemic lupus erythematosus (SLE) in women [6,7,8].

Based on the thermal range of the autoantibodies, AIHA can be classified as either warm type (wAIHA), cold type (cold agglutinin disease (CAD)), or a rarer mixed warm and cold type [6,9]. The wAIHA subtype constitutes the majority of the AIHA cases [4,5,7,10].

Primary wAIHA is reported to be slightly less frequent compared to secondary wAIHA, with an assumed distribution of 35–50% of all wAIHA being primary cases [3,11,12]. Primary wAIHA is the most frequent type in paediatric patients [10]. wAIHA may be associated with many underlying disorders, however most frequent are lymphoproliferative cancers, such as chronic lymphocytic leukaemia (CLL) and connective tissue disorders, especially systemic lupus erythematosus (SLE). Other diseases associated with wAIHA are solid cancers, infections, or other chronic inflammatory diseases, such as colitis ulcerosa or Crohn’s disease [3,6,11,12,13,14,15,16,17,18]. In children, the most frequently reported causes for secondary wAIHA are Evans syndrome (ES), autoimmune diseases, and infections [19]. In addition to underlying diseases, both drugs and organ transplantation can also cause secondary wAIHA [7,20].

Compared with primary wAIHA, secondary wAIHA is associated with poor response to available therapy, and the overall survival is reduced [14,21,22]. Therefore, correct classification of wAIHA as primary or secondary is crucial both for management and follow-up. In this review, we provide an update on the epidemiology of primary and secondary wAIHA to assist the management of patients with wAIHA, including age and sex-ratios, and which diseases are most often associated with secondary wAIHA.

## 2. Materials and Methods

### 2.1. Search Strategy

We conducted a literature search using the PubMed database (NIH, National Library of Medicine, Bethesda, ML, USA) on 14 July 2020. PubMed was searched for articles about AIHA published from 1990 to 2020, using the search terms [“AIHA” OR “AUTOIMMUNE HEMOLYTIC ANEMIA”]. Articles written in English or Danish were considered. The search strategy was broad to allow for identification of all relevant studies for the review.

### 2.2. Eligibility Criteria and Study Selection

All studies were independently reviewed by the two authors, ST and DLH. As a first step, all retrieved articles were screened, based on title and abstract. During this process, the following criteria for exclusion were applied: Abstract-only articles, reviews without original data, laboratory studies without clinical information, animal studies, studies including less than 10 patients, and studies including only patients with CAD, Evans syndrome, or drug-induced haemolysis.

We only included studies with unselected wAIHA patients. Therefore selected cohorts of wAIHA subsets, i.e., studies only including patients with primary wAIHA, only secondary wAIHA, or studies that focused on specific associated diagnoses, such as CLL, SLE, etc. were excluded. Inclusion in our study was not conditioned on the age of patients, and therefore studies including only paediatric patients or only adults patients, where included.

All studies, which met our pre-defined criteria, were included for data-extraction performed by ST and DLH. Any conflicts in inclusion of articles and data were resolved by discussion until consensus between the two authors was reached. Discrepancy in the evaluation were mainly the specified reason for exclusion in studies that fulfilled multiple pre-defined exclusion criteria.

### 2.3. Data Extraction and Processing

From each study, we extracted data regarding proportions of primary vs. secondary wAIHA, the type of hospital involved in the diagnostic workup (primary vs. referral), characteristics of the population, and age-sex distribution of patients diagnosed with AIHA, when available.

### 2.4. Statistics

Collected statistics from the included studies were combined to obtain overall information. Age was mostly reported as median with ranges, but in a few studies as mean and confidence interval. In these few cases, we assumed normality and used the mean as proxy for the median in our analyses. Meta-analysis of medians was not possible. Therefore obtained medians were combined as median of medians, with numbers of included patients as frequency weight. Due to these approximations, confidence intervals were not estimated. Proportions of females among children, adults and all included patients, and the proportion of secondary AIHA in primary, referral, and all level hospitals were estimated using meta-analysis techniques for proportions, with the Stata command metaprop [21,23]. 

### 2.5. Data Sharing and PRISMA Statement

The PRISMA statement was used for composing the structure and contents of this review [22]. All collected data are presented in the tables, and analytical files are available upon request to the corresponding author.

## 3. Results

A total of 2635 publications were retrieved from our literature search and qualified for screening (Figure 1). During title and abstract screening, 2459 articles were excluded, leaving 176 articles for full-text review. After full-text review, 27 articles comprising a total of 4311 patients remained and were included in this review study. The most frequent criteria for exclusion was “*wrong patient population*” according to our pre-defined criteria (Figure 1). The majority of the excluded studies only reported selected patient categories, e.g., patients with ES, SLE, etc.

### 3.1. Age Distribution

Nine studies included only paediatric patients, 10 studies included only adult patients, and the remaining 8 studies included all age groups. Detailed age information is presented in Table 1 and Appendix A. Studies including only paediatric patients comprised 492 patients, and the exclusively adult patient studies comprised 654 patients. The remaining 3165 patients derived from studies including all ages. When all studies were combined, the weighted median age was 68.7 years. In the paediatric cohorts, the weighted median age was 3.8 years, and 53 years in the strictly adult cohorts. The weighted median age among the 8 studies including all age groups was also 68.7 years. The reported age span ranged from 1 month to 95 years [24,25]. A graphic depiction of the distribution of the number of patients, and the median age at diagnosis and study is presented in Appendix A.

### 3.2. Sex Distribution

A male predominance was observed in the paediatric cohorts, where 43% (38; 47) of patients were females. In the cohorts including only adults, a female predominance was noted, 61% (57; 65) compared to 71% (69; 71) in the cohorts containing all age groups. When all studies were combined, 66% (65; 68) of patients were female (Table 1 and Table 2, and Figure 2). Meta-analysis results of the proportion of females by age-groups in the studies are shown in Figure 2.

### 3.3. Causes for Secondary wAIHA

Among the total of 4311 patients with wAIHA, secondary wAIHA was described in 2117 patients or 49.1% (95% CI: 47.6; 50.6). A specific cause was described for 1230 patients (Table 1 and Table 2, and Appendix A). The remaining 50.9% were reported to be primary wAIHA. The proportion of secondary wAIHA was highest among the paediatric patients, where 58.1% (53.6; 62.5) were considered secondary wAIHA. In the adult cohorts, 45.1% (41.2; 49.0) were considered secondary. Some patients were reported to have more than one associated disease for secondary wAIHA, for instance wAIHA associated with SLE and ES, or associated with diabetes mellitus and Hodgkins lymphoma [26,38].

From the studies, we compiled a list of 114 different diseases described as the possible or assumed cause of wAIHA. Some of the underlying diseases, such as SLE and ES were mentioned in many studies, while 49 diseases, e.g., DiGeorge syndrome or herpes virus infection was mentioned only in a single case in all of the included studies (Appendix A). Among paediatric patients, the most frequently mentioned causes of secondary wAIHA were infections and ES. In adult patients, the most frequently mentioned causes were SLE and B-cell lymphomas. The overall most frequently reported causes for secondary wAIHA were SLE, ES, other connective tissue diseases, lymphomas (especially non-Hodgkins lymphomas, CLL), and infections (Table 3 and Appendix A). None of the studies reported a systematic, predefined diagnostic program to categorize primary vs. secondary wAIHA.

### 3.4. Level of Health Care Facility for AIHA Diagnostics

Out of the 27 articles included, 21 articles included patients diagnosed through referral centres, and four studies included patients diagnosed through primary centres. One study used nationwide inclusion from both primary and referral centres, and one study did not describe the health care level of patient recruitment (see Table 1).

The proportion of secondary wAIHA in primary diagnostic centres compared to referral centres is shown in Table 1 and Figure 3 In primary centres, 35.3% (29.2; 41.9) of patients were defined as secondary wAIHA, compared with 60.3% (58.0; 62.5) in referral centres. Meta-analysis results of the proportion of secondary wAIHA by location of wAIHA diagnosis are shown in Figure 3.

## 4. Discussion

Our review study shows that wAIHA occurs in all ages, ranging from early childhood to old age, and it is predominantly diagnosed amongst adult females, and in childhood amongst boys. Contrary to earlier reports, our meta-analysis indicates that primary and secondary wAIHA occur equally frequently. A remarkable heterogeneity with more than 100 different causes for secondary wAIHA was evident from the included studies, and the proportion of secondary AIHA was highest at referral centres.

This systematic review and meta-analysis were derived from all available published studies of AIHA patients within the last 30 years. This provided a very large number of included patients and long observation time.

It is a limitation that our included studies displayed some heterogeneity on the definition of AIHA [48]. Thirteen of the included studies did not distinguish between wAIHA, CAD or mixed type AIHA, but used these terms synonymously. Only the remaining 14 of 27 included studies focused exclusively on wAIHA or provided details that allowed analyses of patients with AIHA and CAD separately. Inconsistencies across studies in the reported descriptive statistics also posed a problem. Especially for age at diagnosis, where some studies displayed medians with inconsistent reporting of interquartile range or minimum–maximum range, while others used means with confidence intervals. This diversity in data made a meta-analysis of age at diagnosis impossible.

### 4.1. Causes for Secondary AIHA

Previous studies have reported that 35–50% of all wAIHA are primary wAIHA subtype in line with our estimate of 51% [3,11,12]. Primary wAIHA was more prevalent amongst non-referred patients, who are more often unselected. However, importantly, no study reported on a systematic diagnostic protocol and the diversity in underlying diagnoses as cause for secondary wAIHA was considerable, and therefore the proportion of secondary wAIHA may be inaccurate.

The highest percentage of secondary wAIHA, ranging between 59 and 93%, was found in the paediatric patients, which is in line with most previous studies [20,49,50]. The most frequently mentioned causes for secondary wAIHA in paediatric patients were ES (78.2%) and infection (87.3%).

Eleven of the 27 studies included ES as a cause of secondary wAIHA, even though AIHA is part of the diagnostic criteria for ES [51]. It is therefore debatable whether ES can meaningfully be categorized as a cause of secondary wAIHA. This divergence might lead to an overestimation of the frequency of secondary wAIHA. ES accounts for 156 cases out of 2117 cases of secondary wAIHA. If ES is disregarded as a cause for secondary wAIHA in this study, the proportion of secondary wAIHA would decrease considerably.

Detailed information regarding causes of secondary wAIHA was available from 26 out of 27 of the included studies. Consistent with previous studies, SLE was the predominant cause of secondary wAIHA in adult patients, comprising 18% of the patients with a defined cause [18]. Further, three studies described “Connective tissue disease” as a cause of secondary wAIHA [42,44,47].

CLL is a frequently reported cause of secondary wAIHA. However, in our review, only 38 such cases were mentioned in the included studies and only among adult patients [18,52,53]. Other haematological malignancies were frequently mentioned in the included studies, with non-Hodgkin lymphomas as the underlying cause in 59 cases. Our study does not confirm the indication of some previous studies that CLL and lymphomas should account for half of all secondary wAIHA cases [6,10].

A predefined list of diseases that defined the wAIHA as secondary was reported in nine of 27 studies. A predefined list may underestimate the proportion of patients with secondary wAIHA, as other causally associated diseases may not be recorded. However, without pre-defined criteria, co-occurring diseases may reflect a random association with wAIHA [37]. For example, many potentially causal diseases are only mentioned in just one or two cases, emphasizing both the heterogeneity in classification, several potential risk factors for developing secondary wAIHA, and the risk of spurious associations. Furthermore, none of the studies reported a systematic diagnostic procedure for classifying wAIHA subtype correctly, which may lead to either a decreased or increased proportion of secondary wAIHA.

A total of 114 different diseases and syndromes were mentioned as a cause of secondary wAIHA. However, 49 of these were only mentioned in one patient in one study (see Appendix A). To our knowledge, no consensus guidelines define which co-occurring diseases classify wAIHA as secondary [49,50]. However, such definitions would allow a more accurate classification and possibly improve future research and clinical care.

### 4.2. Age and Sex Distribution

The median age at wAIHA diagnosis varies depending on the study populations, where eight studies included all ages, nine only paediatric patients, and the remaining 10 studies only included adult patients. Based on all 27 studies with 4311 patients, we estimated the median age of diagnosis to be 68.7 years. Previously, the incidence of wAIHA has been reported to increase after 40 years of age, with a peak incidence between 60 and 70 years, which is in line with our findings [4,5]. However, when we focus on the adult cohorts, the weighted median age was 53 years due to the weight of the “all age” studies, where the majority included more than 100 patients. For children, a peak in wAIHA incidence at around four years of age has been reported, much like our median age of 3.8 years [6,19].

Earlier studies have described a female predominance of approximately 60%, much like our overall estimate of 66% (65; 68) [6,9,54]. The reason for our higher proportion of females, compared with other studies, could be assigned to our low percentage of patients with secondary wAIHA due to CLL, as males have a higher risk of developing CLL [55]. Of note, 57% of paediatric wAIHA patients were boys. The same difference has been seen in other autoimmune haematological diseases, such as immune thrombocytopenia (ITP), where most of the younger adult patients are female, while most of the paediatric patients are boys [52].

### 4.3. Level of Health Care Facility at wAIHA Diagnosis

The majority of the studies included patients through referral centres, which can affect the perceived distribution of primary and secondary wAIHA. We found a large difference in the proportion of patients diagnosed with secondary wAIHA between primary and referral centres of 35% vs. 60%. Further, the largest study used nation-wide inclusion and reported 41% of the included patients to have secondary wAIHA [4]. This difference in proportion of secondary AIHA could be due to more thorough examination at referral centres, but could also be attributed to a referral bias, where the patients with primary wAIHA and a good response to first line treatment, have a low probability of being referred. In oncology, a discrepancy between cancer diagnostics in primary and referral healthcare facilities have been pointed out, and the same may apply to diagnostics of wAIHA [53]. Primary centres and referral centres often have different roles and diagnostic possibilities in the diagnostic workup of cancer, and rare diseases, such as wAIHA, and the diagnostic pathway to diagnosis can vary, depending on the presentation of the disease [53,56]. For example, one study describes the difficulties of diagnosis of cancer in the primary setting, as symptoms that could suggest cancer are common in the primary healthcare facilities, while cancer is relatively rare [53]. The same might apply to diagnostics of wAIHA, as some of the symptoms are relatively common in primary healthcare facilities, however wAIHA is rarely the cause. Like patients with cancer, patients with wAIHA have varying clinical pictures, all which might complicate the diagnostics at the primary healthcare facilities [13]. Additionally, patients who are referred to referral centres may already be selected for a more complicated clinical course, which is more frequent in some secondary wAIHA [49]. Another reason for the higher proportion of secondary cases of wAIHA in referral centres, might be that patients with secondary wAIHA may need specialized and individualized treatment and possible treatment for the underlying disease, which are available in specialized units [49,54].

## 5. Conclusions

This review consolidates previously reported gender distribution with a large female predominance amongst adults and a male predominance amongst children. Further, our overall median age estimate of 68.7 years, is close to previous reports.

We found a lower proportion of secondary wAIHA compared with previous studies. Systemic lupus erythematosus was the single most predominant cause of secondary wAIHA, but causes were numerous, which emphasizes the need for a systematic way of evaluating the relationship between wAIHA and the presumed underlying cause. The proportion of secondary wAIHA in referral centres was 60%, but only 35% amongst patients from primary health facilities, indicating a skewness in cohorts from referral centres, either through more thorough diagnostic workup or from a referral bias, where patients with secondary wAIHA are more likely to be referred.

## Figures and Tables

**Figure 1 jcm-10-01244-f001:**
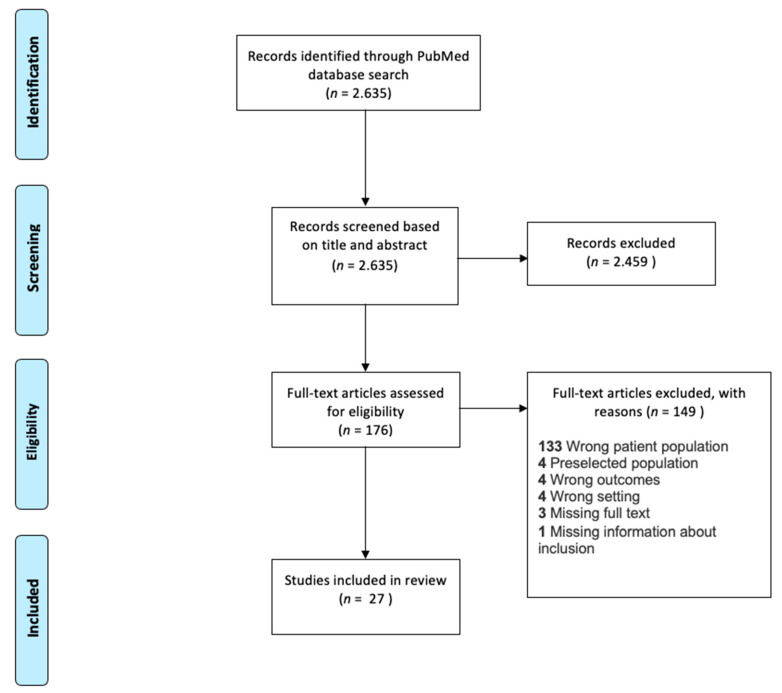
PRISMA flowchart. Studies identified through PubMed 14 July 2020. Studies were first reviewed based on title and abstract, which excluded 2459 studies. 176 studies were eligible for full-text review, and 149 studies were then excluded, mostly due to wrong patient population. 27 studies were included in this systematic review for data extraction and meta-analysis.

**Figure 2 jcm-10-01244-f002:**
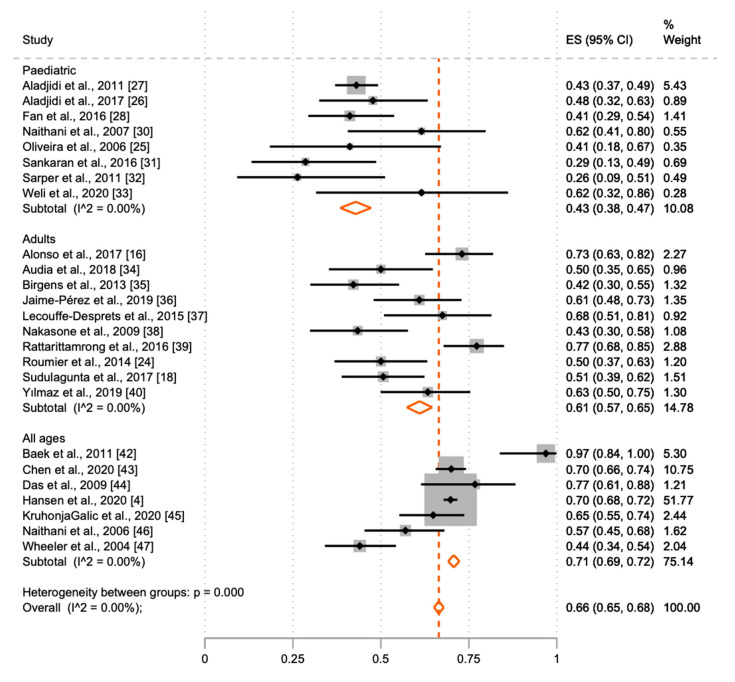
Meta-analysis of the proportion of females in patients with warm autoimmune haemolytic anaemia (wAIHA) with 95% CI and the weight of the included studies. Meta-analysis of the proportion of females with wAIHA by age-group in the studies, generation using the meta-prop procedure in Stata. I^2^ = 0.0% represents a presumably very low heterogeneity in the included studies, as the 95% CI are overlapping. Each of the black ■-symbols represents the ES and 95% CI for one study. The orange ◊-symbols are the pool effect estimates from subgroups and overall.

**Figure 3 jcm-10-01244-f003:**
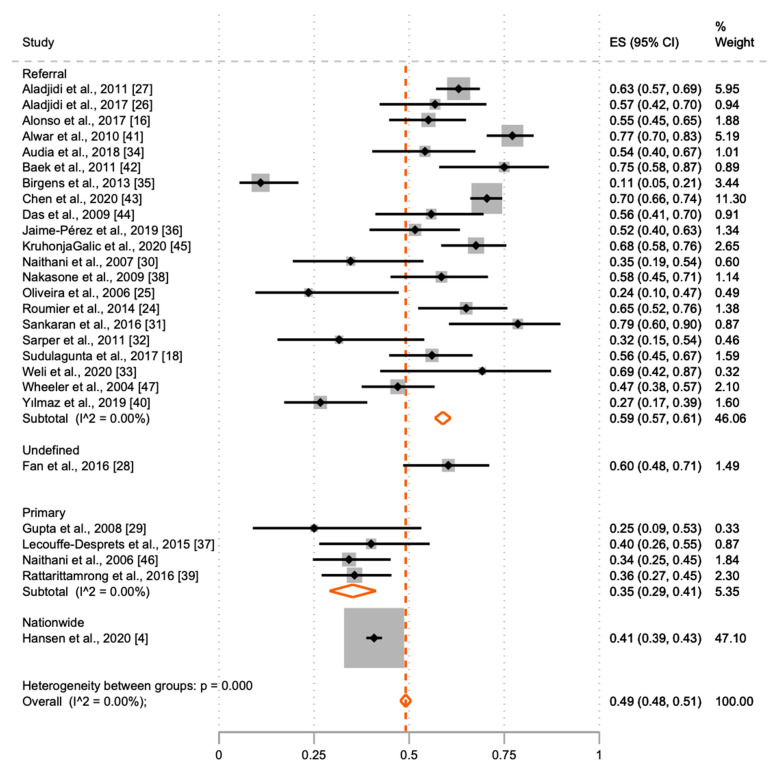
Meta-analysis of proportion of secondary warm autoimmune haemolytic anaemia (wAIHA) with 95% CI and the weight of the included studies. Meta-analysis of the proportion of patients with secondary wAIHA by level of location of wAIHA-diagnosis, generation using the meta-prop procedure in Stata. As shown, more patients are diagnosed with primary wAIHA in primary centres, compared with referral centres. I^2^ = 0.0% represents a presumably very low heterogeneity in the included studies, as the 95% CIs are overlapping. Each of the black ■-symbols represents the ES and 95% CI for one study. The orange ◊-symbols are the pool effect estimates from subgroups and overall.

**Table 1 jcm-10-01244-t001:** Overview of the included studies in a systematic review of patients with warm type autoimmune haemolytic anaemia (wAIHA). The extracted data are number of patients, the female to male ratio, median age and range, number of patients with assumed secondary wAIHA, the age limits of the cohort, and whether the included patients were diagnosed at a primary or referral health care facility.

Study	Number of Patients	Number of Females	Median Age (Range)	Number of Patients with Secondary AIHA	Age Groups	Location of Diagnosis
Aladjidi et al., 2017 [26]	44	21	8.6 (0.1–17.9)	25	Paediatric cohort < 18	Referral
Aladjidi et al., 2011 [27]	265	114	3.8 (0.1–17.4)	167	Paediatric cohort < 18	Referral
Fan et al., 2016 [28]	68	28	1.3 (0.3–15)	41	Paediatric cohort < 16	Not described
Gupta et al., 2008 [29]	12		4.5 (0.6–8)	3	Paediatric cohort	Primary
Naithani et al., 2007 [30]	26	16	11 (2.5–17)	9	Paediatric cohort < 18	Referral
Oliveira et al., 2006 [25]	17	7	0.9 (0.1–15)	4	Paediatric cohort < 15	Referral
Sankaran et al., 2016 [31]	28	8	10 (0.1–18)	22	Paediatric cohort ≤ 18	Referral
Sarper et al., 2011 [32]	19	5	5 (0.1–17)	6	Paediatric cohort < 18	Referral
Weli et al., 2020 [33]	13	8	4.5 (0.2–13)	9	Paediatric cohort	Referral
Alonso et al., 2017 [16]	89	65	36 (17–86)	49	Adult only cohort ≥ 16	Referral
Audia et al., 2018 [34]	48	24	65	26	Adult only cohort ≥ 18	Referral
Birgens et al., 2013 [35]	64	27	66 (35–90)	7	Adult only cohort ≥ 18	Referral
Jaime-Pérez et al., 2019 [36]	64	39	37 (16–77)	33	Adult only cohort ≥ 16	Referral
Lecouffe-Desprets et al., 2015 [37]	40	27	54 (14–86)	16	Adult only cohort > 18	Primary
Nakasone et al., 2009 [38]	53	23	65 (28–88)	31	Adult only cohort	Referral
Rattarittamrong et al., 2016 [39]	101	78	43 (15–83)	36	Adult only cohort > 15	Primary
Roumier et al., 2014 [24]	60	30	53.6 (16–95)	39	Adult only cohort ≥ 16	Referral
Sudulagunta et al., 2017 [18]	75	38	53	42	Adult only cohort ≥ 16	Referral
Yılmaz et al., 2019 [40]	60	38	52 (20–85)	16	Adult only cohort	Referral
Alwar et al., 2010 [41]	175	-	-	135	All ages included	Referral
Baek et al., 2011 [42]	32	31	49 (17–86)	24	All ages included	Referral
Chen et al., 2020 [43]	450	315	51 (31–71)	317	All ages included	Referral
Das et al., 2009 [44]	43	33	31 (12–70)	24	All ages included	Referral
Hansen et al., 2020 [4]	2175	1518	68.7	887	All ages included	All
KruhonjaGalic et al., 2020 [45]	111	72	68 (14–82)	75	All ages included	Referral
Naithani et al., 2006 [46]	79	45	30.5 (0.2–66)	27	All ages included	Primary
Wheeler et al., 2004 [47]	100	44	53.3 (3–90)	47	All ages included	Referral
Total	4311	2654	-	2117		

Alphabetic outline by cohort age characteristics of the included studies with information regarding numbers of patients, number of female patients, the median of the patients, and the number of patients considered having secondary wAIHA. Further information regarding age characteristics of the studies, and where wAIHA was diagnosed is given. Range is provided, if available in the studies. Minimum–maximum age ranges are described in (), when available.

**Table 2 jcm-10-01244-t002:** Summary of data on patients diagnosed with warm type autoimmune haemolytic anaemia (wAIHA) divided into groups according to age characteristics of the studies. The extracted data are number of patients, the female to male ratio, median age in years, and number of patients with assumed secondary wAIHA.

	Patients	Females (%)	Median Age (Year)	Secondary (%)
Paediatric cohorts	492	43.1 (42.5; 43.8)	3.8	58.1 (53.6; 62.5)
Adult cohorts	654	59.5 (58.5; 60.4)	53	45.1 (41.2; 49.0)
All ages	3165	64.4 (64.0; 64.7)	68.7	48.5 (46.8; 50.3)
Total	4311	68.8 (68.6; 69.0)	68.7	49.1 (47.6; 50.6)

A summary of the included studies, divided into exclusively paediatric, exclusively adult, or all ages included, and a total summary of all. A total of 4311 patients were found in the included studies. A total of 69% were female. Note that the female predominance was not found in the exclusively paediatric cohorts. The median age at diagnosis was 68.7 years. In total, 49.1% of the patients had secondary wAIHA due to a variety of diseases.

**Table 3 jcm-10-01244-t003:** The 15 most frequently associated diseases for secondary warm type autoimmune haemolytic anaemia (wAIHA) divided into groups according to age of patients, and the total mentioned in all studies included either only paediatric patients, adult patients or all ages.

Disease	Studies Including Patients of All Ages	Studies Only Including Adult Patients	Studies Only Including Paediatric Patients	Total (*n*)
Systemic lupus erythematosus	51.4%	39.5%	9.1%	220
Evans syndrome	11.5%	10.3%	78.2%	156
Connective tissue disease *	98.9%	1.1%		93
Infection	11.1%	1.6%	87.3%	63
Non-Hodgkin lymphomas	86.4%	13.6%		59
Lymphoid diseases	100%			45
Chronic lymphocytic leukaemia	55.3%	44.7%		38
Unspecified	72.2%	27.8%		36
Autoimmune diseases *	93.8%	3.1%	3.1%	32
B-cell lymphomas		100%		32
Rheumatoid arthritis	46.2%	46.2%	7.6%	26
Antiphospholipid syndrome	14.3%	80.9%	4.8%	21
Diabetes mellitus		90%	10%	20
Renal failure	100%			19
Lymphoma not otherwise specified	55.6%	43.4%		18
Sjøgrens disease	70.6%	29.4%		17

The 15 most frequently mentioned causes for secondary wAIHA out of a total of 114 mentioned causes. For the complete list, see Appendix A. The percentage shows the distribution of one disease or syndrome between the different age groups. * The patients in the “connective tissue disease”- and “autoimmune diseases”-categories are described to be mainly SLE patients [41,45]. The most frequently associated disease in the adult patients is SLE, while the most frequently associated disease in paediatric patients is ES. Unspecified is described in one study (accounting for 15 of the 36 cases) to be drug-induced wAIHA, ITP, or malaria infections [41]. Abbreviations: ES: Evans syndrome, SLE: systemic lupus erythematosus, ITP: immune thrombocytopenia.

## Data Availability

All data used in the systematic review and meta-analysis were available in a publicly accessible repository. The data used in this study are openly available at PubMed, reference numbers [4,16,18,24,25,26,27,28,29,30,31,32,33,34,35,36,37,38,39,40,41,42,43,44,45,46,47].

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
