# Peer review of "Epidemiology of Secondary Warm Autoimmune Haemolytic Anaemia—A Systematic Review and Meta-Analysis"

_jcm, 2021, doi:10.3390/jcm10061244_

Round 1

Reviewer 1 Report

Nice review and impressive amount of work done on available data's.

The main point to be discussed is the definition for secondary AIHA which should be given in the M & M section. Did the authors consider all the diseases listed in Table case as a cause for AIHA? If yes this should be discussed as many of them are not classically considered. 

The proportion of AIHA in children classified as secondary is said to be high but this included post-infection cases: would it be possible to distinguish infection versus other causes of secondary AIHA? Usually associated condition is a poor prognostic factor in this setting but post-infection cases in children have often a limited course. 

The sex distribution is actually difficult to analyse and associated diseases do have an impact. For instance female predominance is obvious in adult patients with SLE but what about CLL population? Is there anyway to address this? And patients with acute infection and patients with  underlying disease are clearly different.

Regarding other causes:

  • I agree that Evans is rather a condition than a cause; especially in children in whom germinal mutations are frequent. Can the authors, as suggested in the discussion section, give the proportion of secondary cases after exclusion of ES cases?
  • the number of CLL cases is low as underlined: for the discussion section would it be possible to have datas on AIAH incidence in published CLL series? This may support a bias for inclusion of CLL cases in AIHA series...
  • many diseases are reported for only 1 patient; discussion should emphasize that in most of the cases this should not be considered as relevant at the moment. Lastly some of the diseases listed in Table 4 may be misleading. For instance what about renal terminal failure: SLE or other AI disease-associated or not? What about diabetes: do you mean only diabetes in young pt? What about prostate cancer? Patient on chemotherapy or not?.../... I agree that these questions may be difficult as limited data's  may be available in original paper but some of the "causes" given should be discussed...
  • Lastly in the discussion section:
    • guidelines on etiological investigations should be referenced and discussed
    • the role of non-diagnosed germinal mutations, which make the patient prone to AI diseases, should be discussed as in the future such investigations will be more and  more frequent

Overall more discussion is needed regarding reported causes.

Tables:

  • Table 1: Authors names should be checked: e.g. Aladjidi and not Aladjini..
  • Table 3: % would be of value as the exact numbers are herein difficult to analyse 
  • Table 4: big amount of work but difficult to read; would it be possible to present this through subgroups e. g. "SLE & other connectivity", "Immune deficiencies", "characterized AI diseases", "malignancies", ...? The exact number of each disease may be given but the landscape of associated disease would be more evident

Figure 3: really useful? To be given only in Suppl. files?

Minor remarks:

  • Non-Hodgkin/Hodgkin lymphoma and not non-Hdgkins

Authors list: to be checked too: e.g. only 1 Aladjidi paper is listed

Reviewer 2 Report

The review and meta-analysis are well planned. However, I note some
aspects: 1) in line 114 "Figure 2" should be eliminated or replaced with Figure 3,
since figure 2 doesn't give age informations;
2) in lines 144-145 the female predominances in adults cohorts, in all age
cohorts and that obtained from combined data do not agree
with the female % values ​​reported in table 2, while the female
predominance
in pediatric cohort agrees with the value in table 2.
3) in line 161 "Figure 3" should be eliminated as it does not describe
the specific causes for secondary wAIHA;
4)in the abbreviations present in line 186 it would be appropriate
to also insert ITP, present in line 184;
5) in line 208 "primary wAIHA" should be "secondary wAIHA";

Reviewer 3 Report

The paper describes data of wAHA found in published paper on that theme in 1990 - 2020 yrs. The authors meticulously analyze the data and discuss them. I think it will be contribution to the knowledge of this ailment described and discussed by hematologists and immunologists.
